# Robust Transfer for Bayesian Optimization with Prior-Data Fitted Networks

## Abstract

Bayesian optimization (BO) is a sample-efficient optimization technique for black-box optimization, and using transfer learning to leverage historical information from related tasks can greatly improve its performance. Multi-task Gaussian processes are commonly used to transfer knowledge from source tasks to target tasks, but these models often make strong assumptions about the relationships between tasks and thus suffer from negative transfer and degraded predictive performance when these assumptions are violated. In this paper, we present Multi-Task Prior-Data Fitted Networks (MTPFNs), a flexible surrogate model that emulates Bayesian inference over user-specified priors over the relationship between tasks. We also propose a novel data-generation procedure specifically designed for the Bayesian optimization transfer setting which enables MTPFNs to be robust to negative transfer and efficiently leverage relevant information. Across a variety of synthetic and real-world benchmarks including hyperparameter optimization, we demonstrate that MTPFNs successfully transfer knowledge in challenging scenarios where existing multi-task Gaussian processes struggle, outperforming existing robust transfer learning methods for Bayesian optimization.

## 1 Introduction

Black-box optimization is widely used in scientific settings and industrial applications to optimize the outputs of resource-intensive processes, especially in scenarios where there is no known analytical form and gradient information is not available. For example, a practitioner may wish to tune the hyperparameters of a machine learning model (Snoek et al., 2012), an engineer may seek to find an optimal design for a new automobile (Liao et al., 2008), and a chemist may aim to design a reaction by choosing the concentrations and experiment conditions (Shields et al., 2021). For these costly design problems, Bayesian optimization (BO) is a popular sample-efficient method that aims to find the global optimum with a minimal number of function evaluations by fitting a probabilistic surrogate model to the observed data and selecting the next evaluation points based on an acquisition function that balances exploration with exploitation.

Because many applications of Bayesian optimization are costly and thus limited to a small number of observations, it can be especially valuable to incorporate information from related tasks. For example, when tuning the hyperparameters for a machine learning model, the performance of previous model evaluations with slightly varied training data or model architectures may offer useful insights. Multi-task Gaussian processes (MTGP) surrogate models are a widely used approach for transferring information from related tasks: by jointly modeling data from the target task with the data from auxiliary sources, the model can leverage the correlations between tasks to improve predictions for the target task. However, this joint modeling introduces a trade-off between data efficiency and robustness. The approaches that fit a single MTGP to all of the data (Bonilla et al., 2007) may make strong assumptions about how different tasks are correlated and experience *negative transfer*, where the inclusion of unrelated tasks degrades performance. Meanwhile, other approaches that fit separate Gaussian Processes (GPs) per task and ensemble their predictions (Tighineanu et al., 2024) are more robust to negative transfer but cannot jointly capture cross-task information. As a result, it is difficult to design surrogate models that simultaneously enable efficient and flexible transfer while remaining robust to spurious task relationships in the classical GP framework.

Prior-data Fitted Networks (PFNs) (Müller et al., 2021) offer an attractive alternative to Gaussian Processes because they are capable of approximating the posterior for any prior over functions that can be sampled from. This enables them to mimic the behavior of Gaussian processes, while also allowing Bayesian inference over complex bespoke priors. However, PFNs have only been applied in single-task settings, where all observations are assumed to come from one underlying function.

In this work, we propose Multi-Task Prior-data Fitted Networks (MTPFNs), a surrogate model explicitly designed for robust transfer learning for Bayesian optimization. We design a novel data generation procedure that mixes data from both related and unrelated tasks, enabling our MTPFN to generalize better in classical transfer learning tasks while being more robust to negative transfer from irrelevant auxiliary sources. This prior would not be easily specified in a GP. Our MTPFN also utilizes a novel hierarchical attention architecture that effectively captures intra-task and inter-task relationships and facilitates transfer. We demonstrate the efficacy of MTPFNs for Bayesian optimization across a diverse set of synthetic problems and real-world hyperparameter optimization benchmarks, which highlights that MTPFNs avoid negative transfer and are more sample-efficient than existing robust transfer learning methods for BO.

## 2 BACKGROUND AND RELATED WORK

### 2.1 BAYESIAN OPTIMIZATION

Bayesian optimization (Garnett, 2023; Frazier, 2018) is a sample-efficient approach for optimizing a black-box function $f$ over a compact domain $\mathcal{X} \subset \mathbb{R}^D$. Often the observations are corrupted by additive noise $y = f(x) + \epsilon(x)$, where $\epsilon(x)$ represents a noise process. At iteration $n$, BO fits a probabilistic surrogate model, typically a Gaussian process (Rasmussen, 2004; Snoek et al., 2012; Lu et al., 2022) or alternatives such as neural networks (Müller et al., 2023; Li et al., 2024; Brunzema et al., 2024), to the collected data $\mathcal{D}_n = \{(x_i, y_i)\}_{i=1}^n$, and yields a posterior distribution $p(f \mid \mathcal{D}_n, \theta)$. An acquisition function $a(x; \theta)$ then uses this posterior to balance exploration and exploitation by quantifying the utility of evaluating a new point (Ament et al., 2025). The next evaluation point is selected by maximizing the acquisition function $x_{n+1} = \arg\max_x a(x; \theta)$. The new observation $(x_{n+1}, y_{n+1})$ is added to the dataset, and the process repeats, iteratively guiding the search toward the global optimum $x^* = \arg\max_{x \in \mathcal{X}} f(x)$. The quality of the surrogate model is key to the success of BO, and improvements in the model typically lead to improvements in performance (Hvarfner et al., 2024).

In the transfer learning setting (Tighineanu et al., 2024; Fan et al., 2022), we aim to find the global optimum of a target task $x^* = \arg\max_{x \in \mathcal{X}} f_0(x)$ while having access to evaluations from auxiliary tasks $\{f_t\}_{t=1}^T$. Each observation consists of an additional task index $t$, and the dataset is $\mathcal{D}_t = \{(x_i, y_i, t_i)\}_{i=1}^{n_t}$. The acquisition function $a(x; \theta)$ is then used to select the next point for the target task. In order for transfer learning to improve BO performance, the surrogate model must be able to infer the relationship between the tasks and use this information effectively.

### 2.2 MULTI-TASK GP SURROGATE MODELS

One approach for transferring knowledge from auxiliary tasks in BO is to jointly model the target and auxiliary data using a Multi-Task (MTGP) (Bonilla et al., 2007; Swersky et al., 2013; Yogatama & Mann, 2014; Poloczek et al., 2017; Joy et al., 2019). The intrinsic coregionalization model (ICM) (Goovaerts, 1997; Swersky et al., 2013) is a widely used MTGP which models each task as a scaled version of a shared latent function

$$f(x, t) = a_t u(x), \qquad u(x) \sim \mathcal{GP}(0, k_{\text{input}}(x, x')).$$

The covariance between input $x$ from task $t$ and input $x'$ from task $t'$ can then be decomposed as

$$\text{Cov}((x, t), (x', t')) = a_t a_t' k_{\text{input}}(x, x') = k_{\text{input}}(x, x') \cdot k_{\text{task}}(t, t'),$$

where $k_{\text{input}}$ is a kernel that represents the covariance between inputs and $k_{\text{task}}$ captures the covariance between tasks. Because all of the tasks share the same input kernel $k_{\text{input}}$, they are forced to share the same functional properties (such as lengthscale). This enables efficient transfer when the tasks share a similar latent structure, but this strong assumption can lead to negative transfer when the tasks have distinct behaviors. There are MTGPs with weaker assumptions, such as the linear model

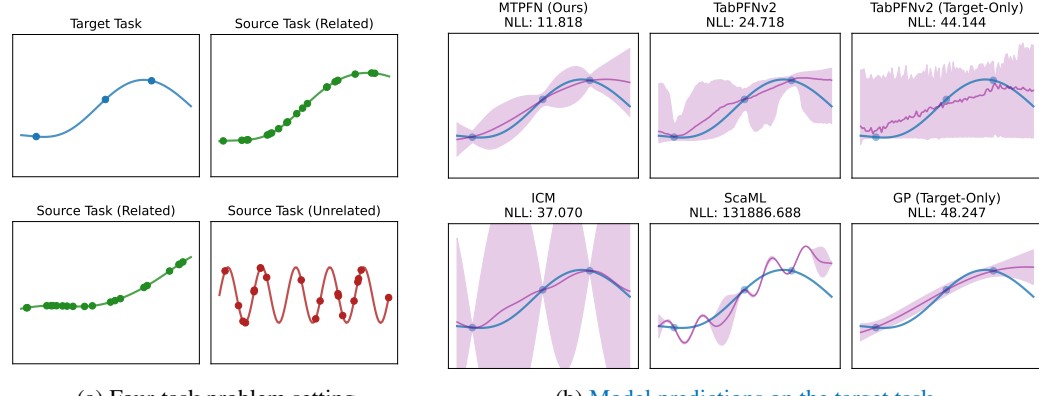

(a) Four-task problem setting  (b) Model predictions on the target task

Figure 1: **MTPFNs effectively transfer information from related tasks while remaining robust to unrelated tasks.** Compared to joint models such as ICM, ScaML, and TabPFNv2 with a categorical task variable, our MTPFN demonstrates improved robustness to the unrelated source task (red). The MTPFN is also able to borrow strength from the related source tasks (green) and outperforms models which only consider the target task. We plot the mean and 95% confidence intervals and compare negative log-likelihoods.

of coregionalization (Goovaerts, 1997), but these flexible models come with higher computational complexity and may lead to overfitting (Alvarez & Lawrence, 2011; Dai et al., 2017). Moreover, MTGPs which jointly model all tasks scale cubically with the number of data points, making them infeasible for transfer learning with many observations across source tasks.

Other approaches focus on scalability: some methods fit separate GPs to each auxiliary task and ensemble their predictions to inform the target prediction (Golovin et al., 2017; Feurer et al., 2018; Wistuba et al., 2018; Dai et al., 2022), but the transfer is less efficient than in an ICM-based MTGP since they do not leverage the same coregionalization assumptions about the similarity between functions. Other methods use data from source tasks to improve the prior distribution over the GP hyperparameters for target-task (Wang et al., 2024; Fan et al., 2022), but this transfer is also less efficient than an ICM-based MTGP because the transfer only occurs through the hyperparameters, and not by correlating observations. Tighineanu et al. (2024) propose a scalable joint modeling approach (ScaML) between the target task and auxiliary tasks. While this improves over per-task GPs by modeling the target and auxiliary data together, it does not capture correlations among auxiliary tasks, leaving a substantial portion of the multi-task structure unexploited. In contrast, our method jointly models all tasks and their interactions in a scalable and robust manner.

### 2.3 Neural Network Meta-Learning

There has been a growing interest in using neural-network based approaches for Bayesian optimization. Single-task methods like OptFormer (Chen et al., 2022) use transformers trained on past optimization trajectories to propose promising query points, while NAPs (Maraval et al., 2023) are trained to predict acquisition function values from large offline datasets. These models can efficiently guide optimization, but generally require substantial domain-specific data during training.

This meta-learning approach has also been extended to multiple tasks: Multi-Task Neural Processes (Kim et al., 2022) require supervised pre-training on large multi-task datasets to learn specific relationships (e.g., weather patterns across cities), limiting them to their training domain. In contrast, MTPFNs perform transfer learning through in-context learning, allowing a single model to dynamically adapt to any configuration of task relationships. Our approach does not have any dependence on the availability of pre-training data, and so our method is particularly suited for Bayesian optimization scenarios where rapid adaptation to new problem contexts given limited data is essential.

### 2.4 In-Context Learning

Transformer neural processes (TNPs) (Nguyen & Grover, 2022) and prior-data fitted networks (PFNs) (Müller et al., 2021) are transformers trained to approximate the posterior predictive dis-

tribution for a prior specified over a hypothesis space $\mathcal{H}$. A PFN, denoted by $f_\theta$, inputs a dataset $\mathcal{D}$ and test point $x_{\text{test}}$ and outputs a distribution over the target variable $p(y_{\text{test}}|x_{\text{test}}, \mathcal{D})$. To train $f_\theta$ to approximate the posterior predictive distribution, we repeatedly sample datasets by first sampling a hypothesis $h \sim p(h)$ which defines a datasets' input-output relationship, and then sampling a dataset $\mathcal{D} \sim p(\mathcal{D}|h)$. The PFN parameters $\theta$ are optimized by minimizing the negative log-likelihood on held-out test examples across datasets, expressed as $\mathcal{L}_{\text{NLL}} = \mathbb{E}_{\mathcal{D} \sim p(\mathcal{D}|h)}[-\log f_\theta(y_{\text{test}}|x_{\text{test}}, \mathcal{D}_{\text{train}})]$. While TNPs and PFNs have successfully applied to Bayesian optimization in the single-task setting (Müller et al., 2023; Nguyen et al., 2024), there has been no prior work which explores the use of in-context transfer of related tasks to accelerate optimization.

Because transfer learning increases the number of in-context data points, the underlying architecture needs to support significantly longer context windows compared to the single-task setting. Various approaches have been proposed to extend the attention mechanisms in transformers to longer contexts, such as sparse attention (Beltagy et al., 2020; Zaheer et al., 2020), hierarchical attention (Wu et al., 2021; Chalkidis et al., 2022), and others (Katharopoulos et al., 2020; Kitaev et al., 2020). See Zhuang et al. (2023) for a survey of efficient methods.

## 3 METHOD

In this section, we present the Multi-Task Prior-Data Fitted Network (MTPFN), a scalable model that uses in-context learning to transfer relevant knowledge from auxiliary tasks. As discussed in Section 2.4, standard PFNs are trained to approximate the posterior predictive for a single task. MTPFNs extend this framework to multi-task transfer learning settings by enabling the model to leverage shared structures between related tasks. This extension requires several key innovations:

- **Designing a multi-task data generation process.** The behavior of a PFN is dependent on the data generation procedure, and it is important to create a multi-task procedure which balances the flexibility in representing diverse inter-task relationships with structured inductive biases.

- **Extending the architecture to multi-task settings.** The model must be able to represent each task effectively to capture inter-task relationships, so it needs to be able to encode the task information. Multi-task transfer learning typically involves more data than single-task settings, so the PFN needs to scale to larger datasets.

### 3.1 DATA GENERATION PROCESS

PFNs are trained to approximate the posterior of a data generation process (DGP), and the design of this prior significantly influences predictive performance. While various DGPs have been proposed in previous works (e.g. Adriaensen et al., 2023; Hollmann et al., 2025; Müller et al., 2023), the transfer learning setting poses unique challenges: there are complex relationships between tasks where information may transfer through shared latent structures, and real-world scenarios frequently contain unrelated tasks that should be ignored. To address these challenges, we propose a transfer learning DGP that learns complex relationships while mitigating corruption from irrelevant tasks.

Our approach, detailed in Algorithm 1, combines two key insights. (1) When tasks are truly related, strong transfer can be achieved by sharing statistical properties like lengthscales across tasks. Therefore, we base our synthetic data generation procedure by sampling from an isotropic ICM to encourage high levels of shared structure. (2) However, since we cannot know a priori which auxiliary tasks will be helpful for modeling the target task, the model must explicitly decide if tasks are relevant. We incorporate this into our data-generation procedure by introducing a probability $p \in [0, 1]$ that an auxiliary task is unrelated and should instead be modeled independently. This DGP encodes the strong inductive bias of shared structure while maintaining the flexibility to differentiate relevant from irrelevant tasks, enabling the PFN to learn a wide variety of inter-task relationships.

In Figure 1, we show that ICM-based MTGPs and existing PFN priors perform poorly in the setting where there is an unrelated task. In contrast, when trained with our robust prior, our MTPFN does not experience negative transfer and accurately mirrors the true behavior, demonstrating the benefit of our robust prior data generation procedure. We present additional discussion and ablations for alternative DGPs in Section A.

**Algorithm 1** Robust Data Generation

---

**Require:** Sequence length $n$, number of tasks $T$, unrelated task probability $p$
    ▷ **Input Sampling and Task Assignment**
1: Sample inputs $\{x_i\}_{i=1}^n \sim \text{Uniform}([0,1]^d)$
2: Sample task proportions $\pi \sim \text{Dirichlet}(\alpha)$
3: **for** $i = 1$ to $n$ **do**
4:     Sample task ID $t_i \sim \text{Categorical}(\pi)$
5: **end for**
    ▷ **Isotropic ICM Covariance Structure**
6: Sample task covariance matrix $K_T \sim \text{LKJ}(\eta = 1)$
7: Sample input lengthscale $\ell \sim \text{Gamma}(3, 6)$
8: Define input covariance $K_X$ on $\{x_i\}_{i=1}^n$ as RBF kernel with lengthscale $\ell$
9: Compute full ICM kernel $K = K_T \otimes K_X$
10: Sample $y \sim \mathcal{N}(0, K)$
    ▷ **Sampling Unrelated Tasks**
11: **for** each source task $j$ **do**
12:     With probability $p$:
13:         Sample new lengthscale $\ell_j \sim \text{Gamma}(3, 6)$
14:         Define RBF kernel $K_X^{(j)}$ on $\{x_i : t_i = j\}$ using $\ell_j$
15:         Resample $y^{(j)} \sim \mathcal{N}(0, K_X^{(j)})$
16: **end for**

---

## 3.2 TASK REPRESENTATION

To facilitate transfer in the MTPFN, it is important to consider how the task itself should be encoded. This encoding can influence how the model integrates information from the various sources and impact its ability to distinguish between helpful and irrelevant tasks.

A natural starting point for representing the task information is to use static, index-based encodings. Specifically, for input $(\mathbf{x_i}, y_i)$, its associated task $t_i \in \{1, \ldots, T\}$ is mapped to a vector representation. This can be done by representing the task as a *categorical feature* $\mathbf{1}_{t_i} \in \{0, 1\}^T$, and the final input is formed by concatenating the original feature vector with the one-hot encoding to form the new input $([\mathbf{x}_i; \mathbf{1}_{t_i}], y_i)$. Another approach uses *learned task embeddings*, where a continuous embedding vector $\mathbf{e}_{t_i} \in \mathbb{R}^d$ is added to the feature space, $\mathbf{z}'_i = \phi(\mathbf{x}_i, y_i) + \mathbf{e}_{t_i}$. While straightforward to implement, these index-based strategies have significant limitations:

- **The maximum number of tasks $T$ must be specified during training.** The model is unable to generalize to a larger number of tasks since it only uses $T$ different encodings, so there is no way to represent a $(T + 1)$-th task during inference.

- **The representation of the task does not depend on the data.** Because both one-hot vectors and learned embeddings initialize the task encodings independently of the data, the model is not able to encode useful semantic relationships within the task embeddings.

- **The model does not scale effectively to additional tasks and observations.** If we perform a standard attention using all of the observations from all of the tasks, the model has a complexity of $O(D^2 T^2)$, where $D$ represents the number of observations per task. This may become prohibitively expensive as we increase the number of tasks or observations.

### 3.2.1 HIERARCHICAL ATTENTION MECHANISM

To address these limitations, we propose a novel scalable attention mechanism for PFNs that effectively leverages the natural hierarchical structure target and source datasets, as shown in Figure 2. Our approach applies hierarchical attention (Wu et al., 2021) to the transfer learning setting to effectively model multi-task data. We separate the computation into two parts: intra-task processing (within each task) and inter-task processing (across tasks).

We start by extracting inter-task information. Each task has its own set of $D_t$ observations. We introduce a learnable [TASK] token, and this acts as a summary for this task: it is updated based on the points within the task, and it will eventually represent the information about what the model

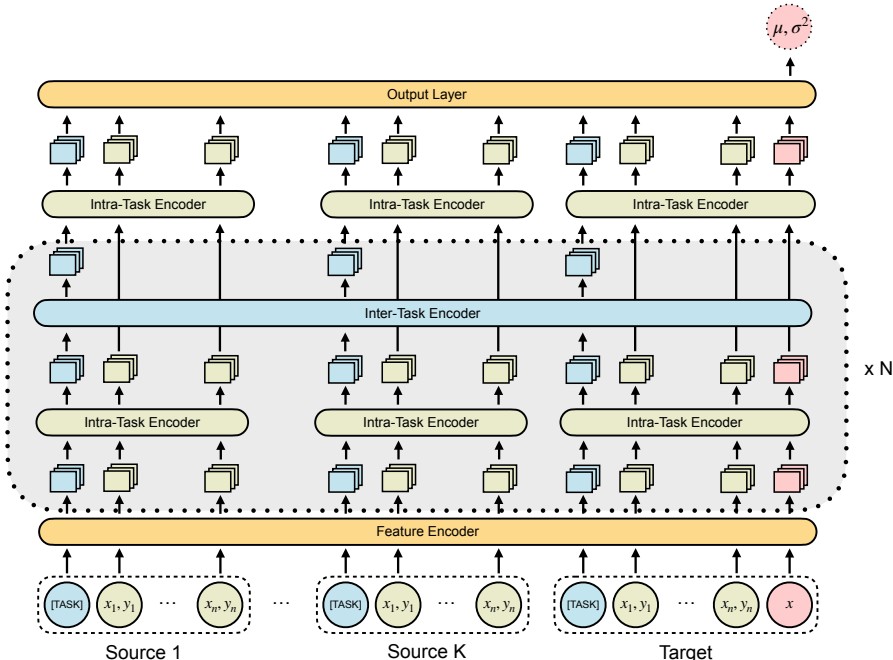

Figure 2: MTPFNs use hierarchical attention and jointly model data across information sources.

has learned about this task. We run a transformer block, the "Intra-Task Encoder", over all of the data points inside this task (plus the [TASK] token). This encoder block learns how the individual observations relate to each other, and it also learns how to effectively update the [TASK] token so it captures a summary of the task's overall properties. The attention is restricted to the observations within a single task and requires $O(D^2)$ compute per task, or $O(D^2T)$ total.

Once each task has an updated [TASK] token, we can use these task summaries to learn relationships across tasks. We feed the [TASK] tokens to an "Inter-Task Encoder", which lets the model learn how tasks relate to one another. The model can also refine each [TASK] token with relevant information from other tasks. Importantly, these encoder blocks only attend to the single summary token per task, not at all of the $D \times T$ observations, so the compute is limited to $O(T^2)$.

For our setup, we interleave the intra-task and inter-task blocks, so the model alternates between "updating each task's internal representations" and "sharing information between tasks." Prior work Chalkidis et al. (2022) has shown that other topologies, such as stacking multiple layers of intra-task blocks before stacking multiple layers of inter-task blocks, may also be effective.

Our hierarchical attention directly addresses many of the limitations of other task encoders. First, our attention mechanism naturally handles inputs of varying lengths, allowing the model to generalize to any number of tasks. This flexibility ensures that even if the model encounters more tasks at test time than it did during pre-training, it can still meaningfully integrate new task representations. Furthermore, our approach enables the model to dynamically learn task representations which depend on the data, and its representation of each task evolves through the many layers of attention. This enables tasks with similar patterns to develop similar representations, allowing the model to better capture the potentially complex relationships between tasks. Finally, the hierarchical structure enables effective scaling, where the computational cost is $O(D^2T + T^2)$ rather than the $O(D^2T^2)$ of standard global attention.

## 4 ADVANTAGES OF MTPFNS

MTPFNs are a compelling surrogate model for transfer learning settings: their flexibility enables them to effectively adapt to diverse, potentially irrelevant source tasks, and they are capable of efficiently scaling to larger datasets than MTGPs. In contrast, although multi-task GPs are commonly

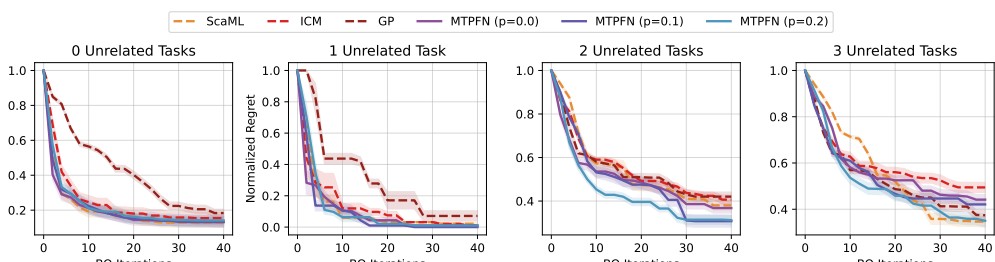

Figure 3: **MTPFNs are robust to negative transfer from unrelated tasks**. We evaluate BO across four multi-task settings, where the target task is related to {0, ..., 3} out of 3 auxiliary tasks. We compare the performance of MTPFNs with different $p$, where $p$ represents the probability that an auxiliary task was drawn independently from the target task during training. As we increase the number of unrelated tasks during evaluation, the MTPFNs which were exposed to unrelated tasks during training ($p > 0$) outperform the ICM model, which suffers from negative transfer. We plot the mean and standard error of the mean over 5 trials.

used for multi-task regression, these models often contain strong assumptions that when violated can lead to negative transfer. Furthermore, they often trade off efficiency with expressiveness: methods which jointly model all tasks capture cross-task interactions, but are computationally expensive, while scalable methods may ignore important inter-task interactions. In this section, we provide explicit demonstrations of the strengths of MTPFNs for transfer learning.

## 4.1 MTPFNs ARE ROBUST TO NEGATIVE TRANSFER

For GPs, the lengthscales are important hyperparameters that control how sensitive the covariance is to changes in the inputs. When modeling multiple tasks, it is often assumed that these tasks all share the same lengthscales (implied by the ICM model). However, this behavior may not be true in practice, and GPs with the ICM kernel may fail to accurately model the problem and suffer from negative transfer, where the inclusion of information from one task hurts the performance on another. In contrast, the flexibility of MTPFNs allow us to train them in a way that explicitly reduces the impacts of negative transfer, as explained in Section 3.1.

In Figure 3, we evaluate the performance of MTPFNs trained with varying proportions of unrelated tasks. In this evaluation setting, there are three auxiliary tasks, where a fixed number are unrelated to the target task. When only one of the auxiliary tasks is unrelated, we find that all of the transfer learning methods perform similarly. However, as we increase the number of unrelated auxiliary tasks to two, we find that the MTPFNs trained on data with a higher proportion of corrupted tasks outperform the ICM-based MTGP, which is more sensitive to negative transfer. When we increase the number of unrelated auxiliary tasks to three out of three, we find that the MTPFN trained with $p = 0.2$ is comparable to the single-task GP, which is the underlying DGP for this problem.

## 4.2 MTPFNs EFFICIENTLY MODEL INTER-TASK RELATIONSHIPS

Many existing Gaussian process surrogate models trade off modeling inter-task relationships with efficiency. To demonstrate the capabilities of MTPFNs, we design a synthetic regression problem with multiple source tasks to highlight the importance of joint modeling. In this setting, all of the data points across all source tasks are drawn from the same function, and this function is highly correlated with the target task. However, there are regions of the input domain where the relevant source tasks do not have any overlap with the target task. Therefore, the model will only be able to make accurate predictions if it is able to leverage the relationship *between* source tasks.

In Figure 4 (Left), we visualize the predictive distributions of the MTPFN, ICM-based MTGP, and ScaML. We see it is necessary to jointly model the target task along with all of the source tasks, as done by MTPFNs and ICM, in order to accurately predict the behavior of the target task across the entire domain. In contrast, ensemble methods such as ScaML, which do not model the joint interactions between auxiliary tasks, are unable to capture the relevant information to make accurate predictions and therefore have less efficient transfer.

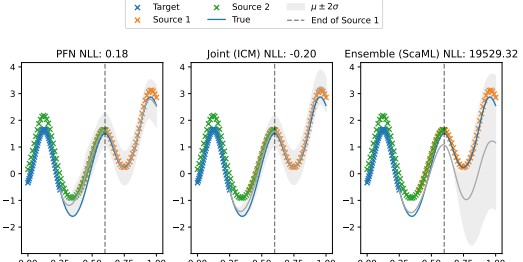
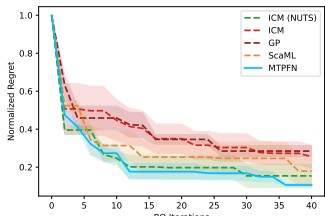

Figure 4: **MTPFNs jointly model the data from the target and all auxiliary tasks and perform fully Bayesian inference**. **(Left)**: MTPFNs perform similarly to other joint models like ICM and outperform ensemble-based models like ScaML. **(Right)**: MTPFNs have comparable performance to fully Bayesian methods like ICM with MCMC (NUTS) sampling.

Although powerful, traditional joint modeling methods like ICM-based MTGPs are unable to scale to a large number of tasks and data points. In Figure 9, we benchmark the runtimes of multi-task models as we increase the number of tasks and the number of data points per task, and we find that this problem quickly becomes unmanageable for ICM-based MTGPs. In contrast, MTPFNs are able to scale to large amounts of data while jointly modeling all interactions.

### 4.3 MTPFNs QUICKLY PERFORM FULLY BAYESIAN INFERENCE

Müller et al. (2021) demonstrate that transformers that are trained to minimize the negative log-likelihood over held-out data from a data-generating process naturally perform Bayesian inference by implicitly learning the posterior predictive distribution. Specifically, the final model outputs a posterior predictive distribution that marginalizes over all of the possible samples from the prior that are consistent with the observed data.

MTPFNs also perform fully Bayesian inference as demonstrated in Figure 4 (Right), where we use an ICM to generate 5 different transfer learning datasets with 3 input dimensions, each with 2 samples from the target task and 20 samples from each of the 3 auxiliary tasks. The auxiliary tasks have varying amounts of correlations with the target task. We then perform 10 runs of Bayesian inference for each transfer learning dataset and summarize the results. We find that MTPFNs perform comparably to the MTGP that uses fully Bayesian inference, which we fit using MCMC sampling through NUTS. Fully Bayesian inference is particularly helpful in the setting where there are very few observations per task and thus there should be high uncertainty over the true inter-task covariance. The ICM model with MAP estimation does not account for this uncertainty and under-performs in this setting. A key advantage of the MTPFN compared to the fully Bayesian MTGP is that the MTPFN is able to make predictions using one forward pass of the model, while NUTS sampling takes significantly longer. See Section B for more detailed timings.

### 4.4 MTPFNs CAN LEVERAGE DOMAIN DATA

When making predictions with PFNs, there are various methods to incorporate domain data to improve the performance. One approach is fine-tuning, where the parameters of a base model are updated to adapt to the specific characteristics of the target domain. This method enables the PFN to specialize to the particular domain; however, fine-tuning is computationally expensive and requires updating the model weights. Furthermore, this method is sensitive to training hyperparameters such as the amount of data and learning rate, and it is also possible to overfit and hurt generalization. Alternatively, these additional sources of data can be provided in an in-context manner to an MTPFN. In this setting, the models are exposed to general transfer learning dataset during training, allowing the model to learn patterns across tasks. During inference, the model uses in-context learning to make predictions and utilizes the auxiliary information without the need for parameter updates.

We demonstrate the benefits of in-context learning compared to fine-tuning on the HPOBench dataset (Eggensperger et al., 2021) for Logistic Regression (LR), which contains 25 tasks with 4 held out for evaluation. We compare three approaches: (1) the "Original Single-Task", a general-

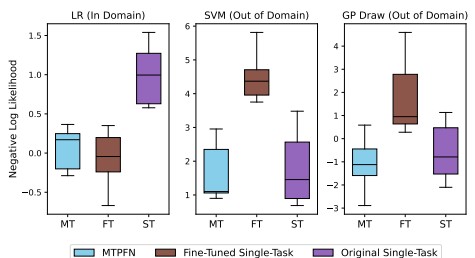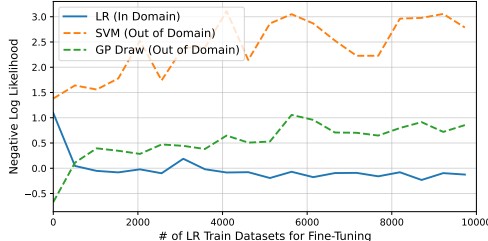

Figure 5: **MTPFNs, which use domain data through in-context learning, match the performance of Fine-Tuned PFNs on in-domain data while generalizing better to other domains.** (**Left**): MTPFNs have comparable NLLs to Fine-Tuned PFNs on in-domain data (LR) and outperform Fine-Tuned PFNs on other domains (SVM and GP Draws). (**Right**): As we fine-tune on more in-domain data, the NLL for Fine-Tuned PFNs significantly worsens for other domains.

purpose single-task model trained on Gaussian process draws with an RBF kernel; (2) the "Fine-Tuned Single-Task", the same base model after fine-tuning on data from the 25 LR training tasks; and (3) "MTPFNs", our method that uses only the 4 hold-out tasks in-context during inference, without any fine-tuning on the 25 training tasks. We evaluate all models by measuring negative log likelihood on the LR evaluation set as well as on other domains (SVM hyperparameter optimization and GP draws) to assess generalization.

The results clearly demonstrate the benefits of MTPFNs over fine-tuned approaches. While fine-tuning does improve performance on the target LR domain, it comes at a severe cost to generalization: as we fine-tune on more LR samples, the performance on SVM and GP domains deteriorates significantly due to overfitting. In contrast, MTPFNs, which use in-context learning, achieve comparable performance on the target domain while maintaining strong generalization across all evaluated datasets. This approach is also computationally efficient, requiring no parameter updates.

## 5 OPTIMIZATION BENCHMARKS

We demonstrate the effectiveness of MTPFNs across various transfer learning tasks for machine learning hyper-optimization. We show that the models are able to effectively utilize domain data while remaining robust to negative transfer in the context of Bayesian optimization.

For our empirical results, we use a transformer backbone with 23 attention layers, where twelve intra-task attention layers are interwoven between eleven inter-task layers. Each attention layer has 4 attention heads with a hidden size of 512. The model is trained on approximately 50 million synthetically generated datasets as described in Section 3.1, with a batch size of 16 and AdamW with a learning rate of 1e-4 and cosine annealing.

We compare our method, MTPFN, to several baselines: (1) ICM (Goovaerts, 1997), a joint method which trains a MTGP on the combined target and source data; (2) ScaML (Tighineanu et al., 2024), an target-aware ensemble method that fits individual GPs to each source task; and (3) a single-task GP which only uses the target task and ignores the source tasks. Our Bayesian optimization results were implemented using BoTorch (Balandat et al., 2020) and GPyTorch (Gardner et al., 2018), and we provide access to our code in the supplementary material.

### 5.1 BENCHMARKS

We compare the effectiveness of the methods on a set of hyperparameter optimization problems for machine learning model through HPOBench (Eggensperger et al., 2021), a collection of tabular benchmarks with hyperparameters and their corresponding loss for models in various settings, as well as the popular FC-Net benchmarks from Eggensperger et al. (2021).

Following Tighineanu et al. (2024), we consider the hyperparameter optimization for five types of models: support vector machines (SVM), logistic regression (LR), XGBoost (XGB), neural networks (NN), and random forest (RF). The source and target tasks correspond to the validation loss when the machine learning model is trained and evaluated on different datasets. For each setting,

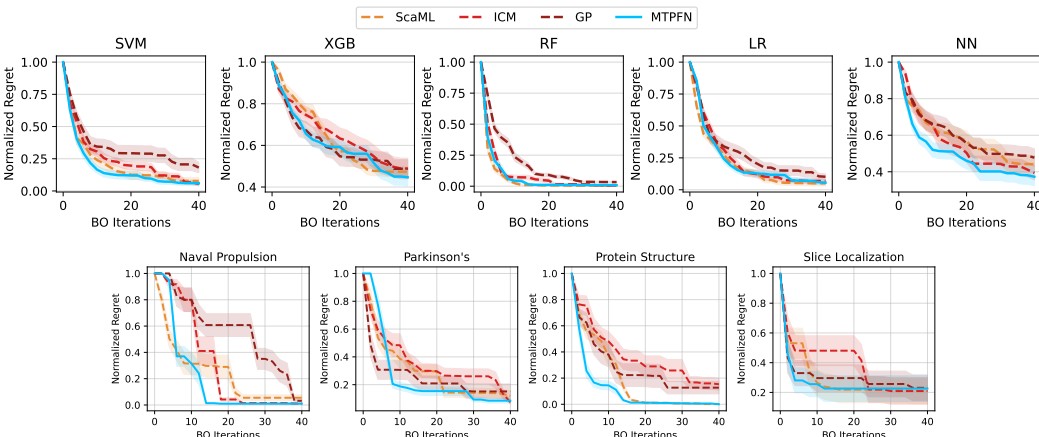

Figure 6: **MTPFNs are competitive across many hyperparameter optimization benchmarks.** Each plot shows the normalized regret for Bayesian optimization loop that was initialized with 3 auxiliary tasks, 20 observations from auxiliary task, and 5 observations from the target task. **(Top):** HPOBench benchmarks **(Bottom):** Tabular FC-Net benchmarks.

we randomly sample one task to be the target function, and we sample 3 source tasks from the remaining. We randomly sample 5 points from the target task and 20 points from each of the sourcetasks to use as the initialization for Bayesian inference. We measure the normalized regret $(f^* - f_{\text{best}})/(f^* - f_0)$ where $f^*$ is the optimal value, $f_{\text{best}}$ is the best value so far, and $f_0$ is the initial value. We run 100 replicates, each with a different combination of target task and source task initializations, and we plot the mean and one standard error.

We share the results of our benchmark in the top panel of Figure 6, and MTPFNs are competitive across all of the model types. Specifically, we find that in instances where the sourcetasks contain helpful information (ScaML and the ICM-based MTGP outperform the GP), the MTPFNs are also able to effectively utilize this information. Furthermore, in cases like XGB where there is negative transfer for the ICM-based MTGP model, we find that MTPFNs are more robust and perform similarly to the standard single-task GP.

We also consider tabular HPO fully connected neural network (FC-Net) benchmarks from HPOBench with different training sets for the FC-Net: Slice Localization, Protein Structure, Naval Propulsion, and Parkinson's Telemonitoring. For each benchmark problem, the goal is to minimize validation loss on the corresponding dataset (target task), using historical data of how the FC-Net performed on the other datasets (source tasks). For instance, the results for Slice Localization use Protein Structure, Naval Propulsion, and Parkinson's Telemonitoring as source data sources. We initialize our Bayesian optimization problem with a random sample of 5 points from the target task and 20 points from each source task. We report the average normalized regret over 20 trials in the bottom panel of Figure 6. MTPFN is the top performer on these problems. This is particularly pronounced when the evaluation budgets are small, which is our setting of interest.

## 6 DISCUSSION

In this work, we present MTPFNs, a scalable and robust surrogate model for Bayesian optimization. By jointly modeling multiple information sources through in-context learning, MTPFNs are able to effectively use historical data and transfer knowledge across tasks. We also introduce a novel data-generation process which enables the model to be more robust to negative transfer, and our empirical results demonstrate that our method is competitive across a diverse set of benchmarks. Our results highlight the effectiveness of leveraging domain data through in-context learning. MTPFNs are able to successfully capture the complex relationships between the information sources and thus can leverage auxiliary information without expensive model-fitting or fine-tuning procedures.

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

## A  DATA GENERATION PROCESSES FOR PFNs

MTPFNs are flexible and have the capacity to incorporate various data generation processes during training. In this section, we explore the impacts of various data generation processes for MTPFNs, each designed to capture distinct inductive biases which improve model performance across different types of tasks.

### A.1  ROBUST ISOTROPIC FULL-RANK ICM

In the main text, we train PFNs with the data generation process described in Algorithm 1: we sample datapoints across tasks from a full-rank isotropic ICM model. This approach assumes that all the input dimensions share identical lengthscales; this assumption imposes a strong prior on the relationship between tasks, and enables effective information transfer across related tasks when the assumption is met. This data generation process enables us to learn information across related tasks, since the full-rank isotropic model makes strong assumptions.

To improve our model's robustness to negative transfer, we also incorporate an additional hyperparameter $p \in [0, 1]$ which dictates the relatedness of the tasks during training. Specifically, $p$ is the probability that any given source task is drawn independently from the target task, and thus may have completely different behaviors and lengthscales. Our data generation procedure enables the model to see a diverse group of datasets which consist of a mix of related and unrelated source tasks.

This $p$ hyperparameter plays a crucial part in the robustness of the model against negative transfer: because the model is able to see many examples of unrelated tasks during training, it becomes more robust to seeing unrelated tasks during inference time and is less likely to be negatively impacted from irrelevant information.

We first introduce inter-task relationships by sampling from an ICM MTGP, where the ICM's assumption of a shared lengthscale across tasks enables strong transfer when the tasks are related. Specifically, we sample an inter-task covariance matrix from an LKJ prior with a concentration of 1.0, which provides us with a diverse set of relationships between tasks, and we sample the shared RBF kernel lengthscale from a Gamma (3, 6) prior following the default lengthscale prior in BoTorch v1.11 (Balandat et al., 2020). To prevent negative transfer, our DGP explicitly encodes the belief that each source task may be irrelevant to the target task by introducing a probability $p \in [0, 1]$ that the task is instead modeled independently using a separate RBF GP with its own lengthscale. In the following sections, we present results under a simple and transparent DGP, but different priors and more sophisticated DGPs can easily be accommodated within the PFN framework. See Appendix A for additional discussions.

In Figure 7, we study the impact of $p$ on the model's ability to accurately predict the empirical data from HPOBench. Specifically, for each model type (SVM, LR, XGB, NN, and RF), we randomly sample one task to be the target task, and we sample 3 auxiliary tasks from the metadata. The target task is randomly initialized with 5 samples, and we also sample 20 points for each of the auxiliary tasks. We measure the mean squared error (MSE) and the negative log-likelihood (NLL) of each surrogate model on heldout examples from the target task, and we repeat this procedure 25 times and plot the average MSE and NLL for each trial.

We find that increasing $p$, which increases the diversity of the data that the model sees during training, leads to improved model performance on real-world benchmarks. We see that the MTPFN trained with $p = 0.2$ consistently outperforms other MTPFNs trained with lower values of $p$, and this MTPFN also outperforms baselines such as the standard ICM model, which assumes that all tasks share the same lengthscale.

### A.2  FULL-RANK ICM WITH AUTOMATIC RELEVANCE DETERMINATION

We can also relax the assumption that all of the input dimensions share the same lengthscale, and instead sample datapoints from an ICM model with Automatic Relevance Determination (ARD), where we assume that each input dimensions has an independent lengthscale. This enables the PFNs to have more flexibility and fit more complex problems; however, this weaker assumption may reduce the model's ability to effective transfer information compared to the isotropic settings.

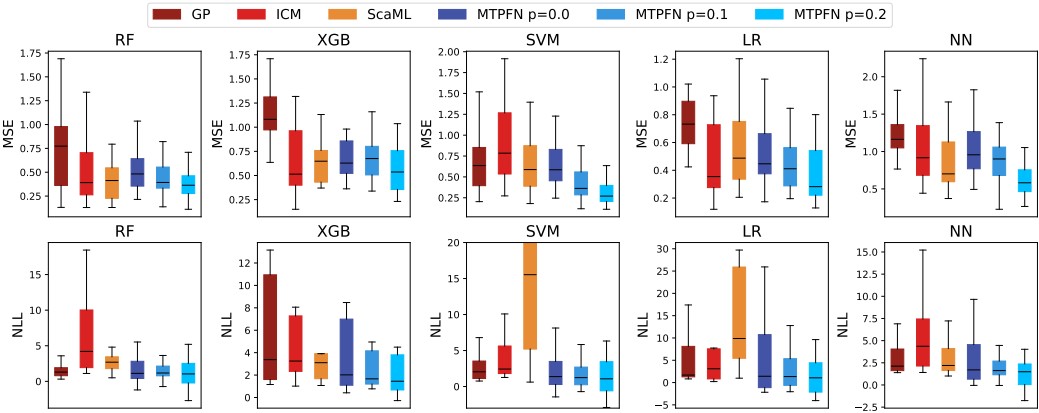

Figure 7: As we increase $p$ (the probability that each source task is unrelated to the target task during data generation), the model becomes more robust to negative transfer and achieves better performance on real-world benchmarks. We visualize the model's predictive performance on the HPOBench dataset, where we sample 5 data points from the target task and 20 data points each from three source tasks. We plot the average MSE and NLL on holdout data from the target task across 25 trials.

---

**Algorithm 2** Data Generation Using a Robust ARD Full-Rank ICM

---

**Require:** Sequence length $n$, number of tasks $T$, unrelated task probability $p$
    ▷ **Input Sampling and Task Assignment**
1: Sample inputs $\{x_i\}_{i=1}^n \sim \text{Uniform}([0,1]^d)$
2: Sample task proportions $\pi \sim \text{Dirichlet}(\alpha)$
3: **for** $i = 1$ to $n$ **do**
4:     Sample task ID $t_i \sim \text{Categorical}(\pi)$
5: **end for**
    ▷ **ARD ICM Covariance Structure**
6: Sample task covariance matrix $K_T \sim \text{LKJ}(\eta = 1)$
7: Sample independent input lengthscales: $\ell = (\ell_1, \ldots, \ell_d) \sim \text{Gamma}(3, 6)^d$
8: Define input covariance $K_X$ on $\{x_i\}_{i=1}^n$ as an RBF kernel with ARD lengthscales $\ell$
9: Compute full ICM kernel $K = K_T \otimes K_X$
10: Sample $y \sim \mathcal{N}(0, K)$
    ▷ **Sampling Unrelated Tasks**
11: **for** each source task $j$ **do**
12:     With probability $p$:
13:         Sample new lengthscale $\ell_j \sim \text{Gamma}(3, 6)$
14:         Define RBF kernel $K_X^{(j)}$ on $\{x_i : t_i = j\}$ using $\ell_j$
15:         Resample $y^{(j)} \sim \mathcal{N}(0, K_X^{(j)})$
16: **end for**

---

We describe this data generation process in Algorithm 2 and highlight the differences from the isotropic data generation in green.

In Figure 8, we compare the performance of the MTPFN trained with isotropic lengthscales (ISO) to the performance of the MTPFNs trained with the ARD lengthscales. This experiment follows an identical setup to Figure 7, where we sample 5 points from a target task and 20 points each from 3 source tasks, and evaluate the MTPFNs on held-out data from the target task. We plot the experiments across 25 trials.

We find that the improved flexibility of the ARD lengthscale generally enables the model to have better performance on the testing data, with the ARD outperforming ISO across many datasets. However, in some settings such as SVM, we find that the model performance of the isotropic ICM and the ARD ICM are comparable. This similar performance may be because the assumption of the

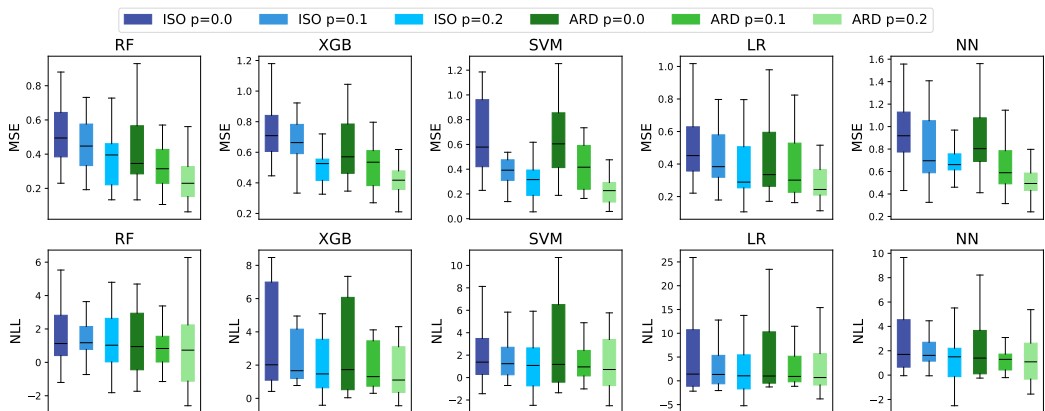

Figure 8: MTPFNs trained with the ARD data generation process tend to outperform MTPFNs trained with the isotropic process (ISO) and achieve lower MSE and NLLs on HPOBench problems.

shared lengthscale across input dimension is satisfied in this setting, so the additional flexibility of the ARD is unnecessary.

# B EMPIRICAL RESULTS AND DETAILS

## B.1 SETUP DETAILS

For our empirical results, we use a transformer backbone with 23 attention layers, where twelve intra-task attention layers are interwoven between eleven inter-task layers. Each attention layer has 4 attention heads with a hidden size of 512. The model is trained on approximately 50 million synthetically generated datasets as described in Section 3.1, with a batch size of 16 and AdamW with a learning rate of 1e-4 and cosine annealing.

We compare our method, MTPFN, to several baselines: (1) ICM (Goovaerts, 1997), a joint method which trains a multi-task GP on the combined target and auxiliary data; (2) ScaML (Tighineanu et al., 2024), an ensemble method that fits individual GPs to each auxiliary task; and (3) a single-task GP which only uses the target task and ignores the auxiliary tasks. Our Bayesian optimization results were implemented using BoTorch (Balandat et al., 2020) and GPyTorch (Gardner et al., 2018), and we provide access to our code in the supplementary.

## B.2 EFFICIENT MODELING OF INTER-TASK RELATIONSHIPS

MTPFNs are able to do Bayesian inference with a single forward pass. We compare the runtime of MTPFNs to joint-modeling methods such as ICM and ensemble-based methods such as ScaML in Figure 9. We see that MTPFNs are able to perform inference on an order of magnitude more data points and tasks compared to traditional GP methods. Furthermore, our proposed hierarchical attention mechanism enables the MTPFN to scale in $O(D^2T + T^2)$, where $D$ is the number of data points per task and $T$ is the number of tasks. In the lower panel of Figure 9, we compare the hierarchical attention to standard global attention, which scales $O(D^2T^2)$. Our proposed architecture significantly improves the efficiency of the MTPFN.

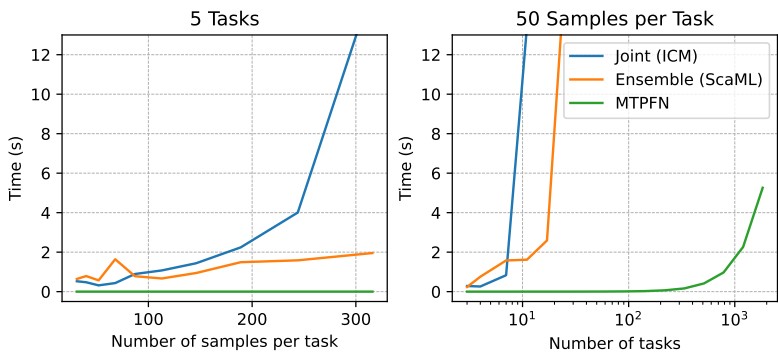

(a) MTPFNs are significantly faster than alternative GP-based methods.

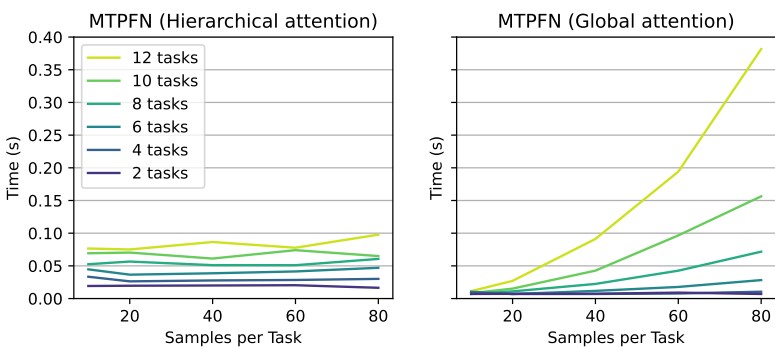

(b) The hierarchical attention mechanism improves the scaling performance of MTPFNs.

Figure 9: Hierarchical MTPFNs are significantly faster than alternative methods.

## B.3 FULLY BAYESIAN INFERENCE

When trained on a data-generation process that draws samples from a multi-task GP with an ICM kernel, we see in Figure 10 that the MTPFN and the MTGP (ICM kernel with MAP estimation) have comparable behavior across varying levels of correlations. Furthermore, in low-data settings demonstrated by Figure 11, we find that the MTPFN outperforms the MTGP because it considers the uncertainty over the task covariance matrix. This demonstrates that fully Bayesian inference may be preferable to MAP estimation.

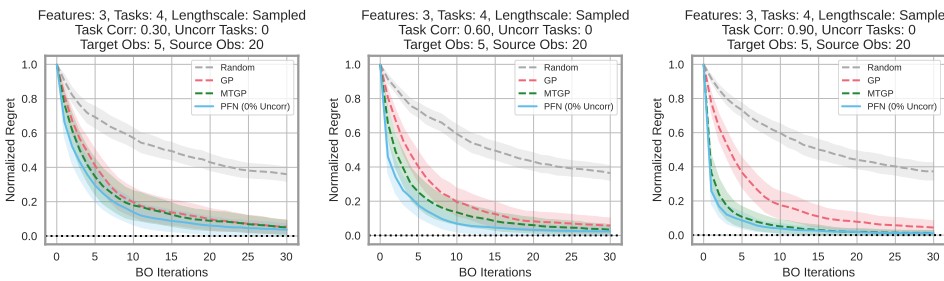

Figure 10: ICM PFNs are comparable to MTGPs across varying levels of correlations between tasks.

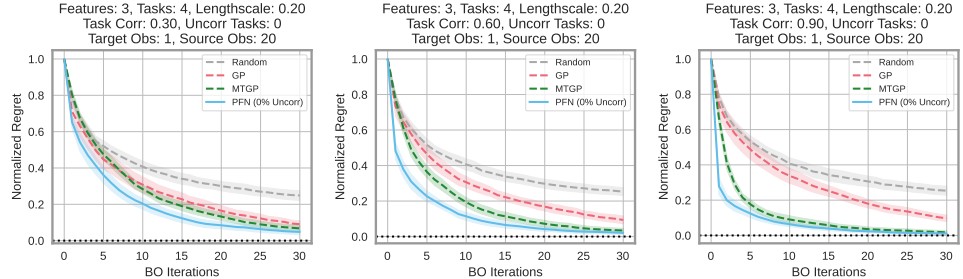

Figure 11: In low-data settings, ICM PFNs, which approximate fully Bayesian inference, outperform MTGPs with MAP estimation. ICM PFNs are comparable to MTGPs across varying levels of correlations between tasks.

## B.4 LEVERAGE DOMAIN DATA

**Original Single-Task PFN Details** The data generation process for the single-task PFN randomly samples inputs $x$ from the unit cube, and then samples the corresponding outputs $y$ by drawing a sample from a GP with an RBF kernel with a lengthscale sampled from Gamma(3, 6). For this experiment, we use a fixed feature size of 2. We train an 8-layer standard transformer (not hierarchical attention) with an embedding size of 256 on this data generation process for 4 million sampled datasets, with a batch size of 16, and AdamW with a learning rate of 1e-4 and cosine annealing.

**Fine-Tuned Single-Task PFN Details** To fine-tune on the LR dataset, we develop a subsampling data-generation procedure: On the 20 training tasks, we subsample within one task to get 50 $x, y$. We uniformly select some number of them to be used as ICL training, and the remaining to be used as the test. We fine-tune the base model described above with a batch size of 16, and AdamW with a learning rate of 1e-4 and cosine annealing.

## B.5 REAL-WORLD OPTIMIZATION PROBLEMS

We consider the hyperparameter optimization for five types of models: support vector machines (SVM), logistic regression (LR), XGBoost (XGB), neural networks (NN), and random forest (RF). For each setting, we randomly sample one task to be the target function, and we sample 3 auxiliary tasks from the meta-data. We randomly sample 5 points from the target task and 20 points from each

of the auxiliary tasks to use as the initialization for Bayesian inference. We measure the normalized regret $(f^* - f_{\text{best}})/(f^* - f_0)$ where $f^*$ is the optimal value, $f_{\text{best}}$ is the best value so far, and $f_0$ is the initial value. We run 100 replicates, each with a different combination of target task and auxiliary task initializations, and we plot the mean and one standard error.

