# OpenReview forum: "Robust Transfer for Bayesian Optimization with Multi-Task Prior-Fitted Networks"
_ICLR.cc/2026/Conference — Submitted to ICLR 2026_

### Official Review · Reviewer_dMBL · 2025-10-24

**Soundness:** 2
**Presentation:** 2
**Contribution:** 2
**Rating:** 2
**Confidence:** 4

**Summary:**

The article introduces a multi-task prior-data fitted network in order to share information across tasks in multi-task Bayesian optimisation. It is deployed and, unsurprisingly, performs better than the baselines on the tasks that are presented.

**Strengths:**

Multi-task Bayesian optimisation is hard, largely due to the standard Gaussian process frameworks either failing, or being too data hungry to learn the between-task correlations (and then too slow when sufficient data is presented). I think that this prior-data fitted network approach is potentially valuable and useful. Performance in the selected tasks seems to be good.

**Weaknesses:**

The description of the actual contribution, the MT-PFN, is very loose. We are given a picture (Fig 2) and very little else. This is the key contribution, and I really wanted to understand it.
The article talks about explicitly accommodating a probability of a task being irrelevant, but I could not work out how this is encoded in the MT-PFN.
There are better methods for testing whether a multi-task model fits multi-task data than slamming it into BO and measuring optimisation performance. Really, this isn't a BO article at all. I would have loved to see some performance metrics for how will the MT-PFN fits data.

**Questions:**

How does MT-PFN actually work? What is the construction, and why does it make sense?

---

> ### Author Response · Authors · 2025-11-24
> **Response [1/2]**
>
> Thank you for the review. We address your points below, and we have also updated the pdf accordingly.
>
> > The description of the actual contribution, the MT-PFN, is very loose. We are given a picture (Fig 2) and very little else.
>
> We respectfully note that the MTPFN is described in detail in the manuscript: Section 2.4 provides the necessary background for general training of PFNs, and Section 3 provides a full description of the two parts of our contribution: the prior data generation procedure, and the hierarchical attention mechanism. Furthermore, Appendix A and B.1 also devote multiple pages to training procedures and implementation details. Figure 2 is only intended as a high-level illustration.
>
> However, inspired by this feedback, we have made significant additions to our methods section in the main text to further detail the core contributions of this work. We hope this helps clarify our paper’s key ideas.
>
> > How does MT-PFN actually work?
>
> A Prior-Data Fitted Network (PFN) is a model trained to approximate the posterior predictive distribution for a synthetic data generation process. The model $f_\theta$ takes a dataset $D = \{x_i, y_i\}$ and a test point $\{x_\text{test}}$ as input and outputs the parameters of a distribution $p(y_\text{test} | x_\text{test}, D)$. The model parameters $\theta$ are optimized by minimizing the negative log-likelihood on test examples across many synthetically generated datasets sampled from the prior. PFNs are trained to perform in-context learning: for each sampled dataset, the input-output pairs are encoded through feature embeddings and fed to a transformer along with the test input embeddings. The PFN then outputs distribution parameters (such as
> $\mu$ and $\sigma$ to parameterize a normal distribution) for each input. We refer you to Section 2.4 for the relevant background, and [1] for additional information about general PFNs.
>
> [1] Müller et al, Transformers Can Do Bayesian Inference
>
> Unlike standard PFNs, which are trained to approximate the posterior predictive using information from only one task, MTPFNs are designed to do transfer learning across multiple tasks. This requires two key innovations:
> 1. **Designing a multi-task data generation process.** Since the behavior of the PFN is dependent on the design of the prior, this is a crucial component of the MTPFN. We discuss the prior design in Section 3.1, explicitly show the procedure Algorithm 1 (moved from the appendix into the main text for the rebuttal), and show its ability to improve against negative transfer in Figure 1 and Figure 4. We also explore variations of the data generation procedure in Appendix A and show its impact on model fit and downstream tasks.
> 2. **Extending the architecture to multi-task settings.** The model must encode the task information and effectively capture inter-task relationships. Furthermore, since transfer learning settings typically require more data than single-task settings, our model needs to scale efficiently to more datapoints. We address the model design updates in Section 3.2, and include new experiments for the rebuttal in Figure 9 to measure the benefits of the our proposed architecture.
>
> > I would have loved to see some performance metrics for how well the MT-PFN fits data.
>
> We provide many examples of model fits throughout the paper:
> - Figure 4 visualizes MTPFN compared to ICM and ScaML fits and reports the NLL
> - Figure 5 provides the NLLs for MTPFNs compared to fine-tuned PFNs
> - Figure 7 and Figure 8 provides the NLLs and MSEs of many different MTPFNs compared to baselines such as single-task GPs, ICM, and ScaML
> - New for the rebuttal, we have also updated Figure 1 to include the NLLs
>
> We consistently find that MTPFNs outperform existing baselines when it comes to model fit, which also explains its improved performance on BO tasks.

---

> > ### Comment · Reviewer_dMBL · 2025-11-27
> >
> > Thank you for your careful response.
> >
> > The new text in Section 3 is a significant improvement in terms of describing what is going on with MTPFN's.
> >
> > I still don't really have an appreciation for when or whether MTPFN's will work with a suite of tasks or otherwise. There is a very specific "In distribution with prob p, out of distribution with prob (1-p)" setup, which feels like it will work only if it matches the data, or you get lucky. I can't see evidence for or against that though. More careful investigation, instead of purely positive claims, would be an improvement.
> >
> > In terms of the figures, I have been reviewing the non-appendices, so in the previous version only Figs 4 and 5 provided any evidence of fit. There is minimal discussion of these results in the text, other than the claim that performance is better than for previous methods. Again, more care to carefully discuss the fits would be an improvement.
> >
> > I will increase my score slightly, since the key description of the MTPFN is now present in the main text.

---

> ### Author Response · Authors · 2025-11-24
> **Response [2/2]**
>
> > There are better methods for testing whether a multi-task model fits multi-task data than slamming it into BO and measuring optimisation performance.
>
> Our work is explicitly framed around transfer learning within BO, not just multi-task learning. While multi-task methods focus on learning the relationships between tasks, the objective of BO is to make efficient decisions under uncertainty. Our data generation procedure is designed for the PFN to identify the patterns which are useful for iterative BO setting, rather than only improving predictive accuracy across multiple tasks.
>
> Furthermore, many papers in BO literature focus on designing effective surrogate models (e.g. [1, 2, 3]). Our work follows the same paradigm, where the PFN is a learned surrogate model trained to produce useful posterior predictive distributions which are then used for sequential decision-making.
>
> [1] Tighineanu et al, Transfer Learning with Gaussian Processes for Bayesian Optimization \
> [2] Feurer et al, Scalable Meta-Learning for Bayesian Optimization using Ranking-Weighted Gaussian Process Ensembles \
> [3] Wistuba et al, Scalable Gaussian process-based transfer surrogates for hyperparameter optimization
>
> We have carefully addressed the points raised in your review, including clarifying several aspects of the method and our empirical results. With these additional details and our additional revisions of the manuscript, we hope you will consider updating your score accordingly.

---

### Official Review · Reviewer_gndA · 2025-10-31

**Soundness:** 2
**Presentation:** 2
**Contribution:** 2
**Rating:** 4
**Confidence:** 3

**Summary:**

The paper proposed a multi-task surrogate model (MTPFN) for bayesian optimization by jointly modeling multi-task data through in-context learning. The proposed MTPFN model is a sophisticated integration of several advanced concepts: prior-data fitted networks (PFNs), in-context learning, and hierarchical attention mechanisms.

**Strengths:**

-	Robustness: Introduces a robust multi-task surrogate model that mitigates negative transfer.
-	Significance: It combines the flexibility of transformer-based models with the robustness needed for real-world multi-task learning.
-	Performance: Demonstrates promising empirical performance across synthetic and real-world benchmarks.
-	Scalability: The hierarchical attention mechanism allows it to handle more tasks and data points than cubic-scaling GP-based joint models.

**Weaknesses:**

-	Incremental Technical Novelty: The core ideas (e.g., hierarchical attention, prior-data fitted network) build heavily on existing literature, making the overall contribution incremental.
-	Unclear Methodological Clarity: The method section lacks a clear illustration of its methodological details (e.g., the problem formulation, the overall procedure of the MTPFN method, and the exact network architecture).
-	Computational Cost of Training: Training the transformer on 50 million synthetic datasets is incredibly computationally intensive.
-	Dependence on Synthetic Prior: Performance is contingent on the quality of the data generation process. A poorly specified prior could lead to suboptimal performance on real-world tasks.

**Questions:**

-	Potential Limitation: What are the limitations of the proposed Multi-Task Prior-Data Fitted Network?
-	Ablation Study: A more explicit ablation study quantifying the individual contribution of the hierarchical attention versus the robust DGP would be valuable.
-	Hyperparamenter Sensitivity: How sensitive is the model's performance to the specific hyperparameters of the DGP (e.g., the Gamma priors for the lengthscale)?
-	Small Synthetic Dataset: Could the model be effectively trained on a smaller number of synthetic datasets, or are the 50 million samples critical for stable performance?

---

> ### Author Response · Authors · 2025-11-24
> **Response [1/2]**
>
> Thank you for your review. We address your points and questions below, and we have also updated the pdf with significantly more detail regarding the related work and our proposed method.
>
> > The core ideas build heavily on existing literature, making the overall contribution incremental.
>
> We respectfully disagree with your assessment. Our contributions go beyond straightforward extensions of prior work, and we are the first to consider PFNs for the transfer learning setting. Prior works focus on training PFNs to perform well in single-task or domain-specific settings, and it is not obvious a priori that these models could effectively learn inter-task relationships. Furthermore, although hierarchical attention has been previously explored in other contexts, our paper is the first to propose it for scaling PFNs to multi-task settings.
>
> Moreover, there have been many papers published to top ML conferences with similar scopes and styles, where the main novelty lies in designing a synthetic prior for a new domain. For example
> - PFNs4BO (ICML 2023) [1] propose a PFN surrogate more single-task BO, and the main contribution of the paper is the synthetic data generation procedure.
> - ForecastPFN (NeurIPS 2023) [2] train a PFN by defining a synthetic prior for time series, and demonstrate its empirical performance compared to other forecasting methods.
> - Other examples include FT-PFN (ICML 2024) for freeze-thaw BO, and NSL-PFN (ICML 2025) for neural scaling law extrapolation.
> Our work follows this paradigm of defining a new synthetic data generation procedure to enable improved performance on real-world downstream tasks, but it applies this to the novel transfer-learning setting and also includes contributions which enable robust transfer and scalability.
>
> [1] Müller et al, PFNs4BO: In-Context Learning for Bayesian Optimization (ICML 2023) \
> [2] Dooley et al, ForecastPFN: Synthetically-Trained Zero-Shot, (NeurIPS 2023) \
> [3] Rakotoarison et al, In-Context Freeze-Thaw Bayesian Optimization for Hyperparameter Optimization (ICML 2024) \
> [4] Lee et al, Bayesian Neural Scaling Law Extrapolation with Prior‑Data Fitted Networks (ICML 2025)
>
> > The method section lacks a clear illustration of its methodological details.
>
> Thanks for your feedback. We have made significant additions to our methods section in the main text to further detail the core contributions of this work. We hope this helps clarify our paper’s key ideas.
>
> > Computational cost of training
>
> As with all PFN-type methods, there is an initial one-time training cost incurred by us. In our case, we trained our models for 24 hours on an NVIDIA V100. However, after this one-time training, the model can be repeatedly applied with very little cost. Furthermore, we will release the model weights, so future practitioners will be able to directly use the model without their own pre-training.
>
> After this initial cost, the inference time of PFNs is significantly faster than GP-based methods. As we demonstrate in Figure 9 in the supplementary material, MTPFNs are able to perform inference on an order of magnitude more data compared to traditional GP methods, with joint-modeling methods such as ICM taking up to 12 seconds to fit 300 samples, ensemble-based methods such as ScaML taking up to 2 seconds, and the MTPFN taking only 0.02 seconds. Furthermore, we see in Figure 4 (right) that MTPFNs are comparable to NUTS sampling; however, the NUTS algorithm can take upwards of 10 minutes per iteration; meanwhile, the MTPFN is able to perform inference in under 1 second (~600x speedup).
>
> > Dependence on synthetic prior
>
> Our empirical results demonstrate **consistent competitive performance across diverse benchmarks**. In our empirical evaluation, we find that MTPFNs are able to confidently leverage relevant information from related tasks while avoiding negative transfer from unrelated ones. Furthermore, although the value of $p$ is fixed during training, through the millions of generated synthetic datasets that the model is trained on, the model still sees many instances where all of the source tasks are unrelated, none of the source tasks are unrelated, and everything in between. With this data generation procedure, we expect the model to adapt fairly well at test-time to any number of unrelated tasks. We also include sensitivity analyses in Appendix A, where we vary the synthetic data generation process and benchmark how different PFNs perform across our diverse set of real-world tasks.

---

> ### Author Response · Authors · 2025-11-24
> **Response [2/2]**
>
> > A more explicit ablation study quantifying the individual contribution of the hierarchical attention versus the robust DGP would be valuable
>
> The hierarchical attention and the robust DGP address two orthogonal aspects of our model: the updated model architecture improves the scalability of the model, while the DGP enhances the predictive distribution. Because these effects are independent of one another, the existing ablation studies in our work are naturally split between demonstrating the benefits of the attention mechanism vs the DGP.
>
> For instance, to better understand the impact of the robust DGP, we explore the impact of different DGPs in Figure 3 as well as Appendix A. We benchmark the performance of MTPFNs trained with varying proportions of unrelated tasks $p$, and we find that increasing $p$ enables the MTPFN to be more robust for downstream applications which have more unrelated tasks. In comparison, MTPFNs trained with $p=0$ (no unrelated tasks) suffer from negative transfer, similarly to GP models such as the ICM.
>
> However, inspired by your review, we have run new experiments (shown in Figure 9) to better demonstrate the contribution of the hierarchical attention mechanism compared to standard global attention. We now show the runtime of the transformer with and without the hierarchical attention, and we see that the hierarchical mechanism enables significantly faster inference as we increase the number of tasks.
>
> > What are the limitations of the proposed work?
>
> The limitations of our approach are largely inherited from the PFN framework itself. As you have identified, PFNs require an up-front training cost compared to traditional surrogate models such as GPs. Furthermore, the success of the PFN is dependent on the choice of synthetic prior, and the question of what data generating procedure is “optimal” remains an open and important research direction. We hope that our paper inspires further exploration in this area. For instance, priors inspired by LMC models could capture more complex inter-task relationships, while priors which incorporate low-rank task relationships could also provide alternative useful inductive biases. We believe our work provides a solid foundation for exploring these richer prior generation procedures.
>
> We have put significant effort into addressing your questions, updating the manuscript for clarity, and running new experiments for the rebuttal. We hope you will consider increasing your score accordingly!

---

### Official Review · Reviewer_f9Ye · 2025-11-06

**Soundness:** 3
**Presentation:** 2
**Contribution:** 3
**Rating:** 6
**Confidence:** 3

**Summary:**

The paper studies the problem of Bayesian optimization in transfer learning setting using Prior Fitted Networks. The authors present a surrogate model called Multi-Task Prior-Data Fitted Networks that emulates Bayesian Inference making it effective to optimize for a new test function by exploiting the relationship between training and test tasks through heirarchical attention mechanism. They also propose a novel data-generation process for training the model that makes it robust against negative transfer. Finally, they also present synthetic and real world experiments to showcase the effectiveness of their method over baselines like MTGP in transfer learning BO domains.

**Strengths:**

1. The paper presents an interesting method for transfer learning in Bayesian Optimization setting using Multitask Prior Fitted Networks.
2. Their model MTPFN is able to adapt test function through incontext samples without requiring retraining which makes it really efficient in practice.
3. The hierarchical attention mechanism introduced in the paper is very interesting as it reduces the computational complexity allowing it to scale efficiently over larger number of tasks and data points.
4. The experiment results demonstrates the effectiveness of MTPFNs to robustly model inter-task relationships and avoid negative transfers.

**Weaknesses:**

1. The paper states the experimental observation that MTPFNs are robust against negative transfers but does not provide much insight into why and when does that happen.
2. Some of the details in the paper appear too condensed. For example, Hierarchical Attention Mechanism discussed in line 240-262 is very dense and it would be really helpful to add sufficient details to make it more comprehensible to readers.
3. The entire MTPFN model is trained to "emulate" the posterior of a specific synthetic Data Generation Process (DGP). If real-world tasks and their relationships do not resemble this mixture prior, the model's performance may degrade significantly.
4. The paper does not address scalability wrt input dimensionality of $\mathcal{X}$.

**Questions:**

1. Section 4.1. mentions MTPFNs are robust against negative transfers but it lacks sufficient explanation or intuition to support the claim. Can you please clarify on that? Is this a totally empirical observations and if so why do you think this would generalize to other examples?
2. The paper demonstrates that a certain value of $p$ improves robustness. But how sensitive is the model's inference-time robustness to this parameter? For instance, if the model is trained with $p=0.2$ but deployed in a scenario with 90% unrelated tasks, will it still perform well, or does this $p$ value need to be "tuned"?
3. In section 4.4, the authors make comparison to fine-tuning approaches and observe that they are brittle. Did they try comparing to more modern PEFT methods, such as LoRA? Since, these methods are designed to adapt to new task without destroying prior knowledge, I suspect they may present a much stronger baseline against MTPFN's in-context approach which would be an interesting comparison to have.

---

> ### Author Response · Authors · 2025-11-24
>
> Thank you for your thoughtful review! We have addressed your comments and questions below, and we have also updated the pdf with significantly more detail regarding the related work and our proposed method.
>
> > The paper states the experimental observation that MTPFNs are robust against negative transfers but does not provide much insight into why.
>
> MTPFNs are not inherently robust to negative transfer. Instead, it takes careful consideration and design to explicitly encourage this behavior, and in fact, providing this capability is one of the main contributions of our paper.
>
> A Prior-Data Fitted Network (PFN) is trained to perform in-context learning, meaning its performance on a new task is entirely dependent on the synthetic data generation procedure that it is trained on. Previous data generation procedures for PFNs (such as PFNs4BO and TabPFN) were not designed for the transfer learning setting and thus do not explicitly encourage robust transfer.
>
> Our novel data generation procedure, in contrast, exposes the PFNs to diverse and even unrelated tasks. This trains the PFN by teaching it to focus on relevant patterns and avoid being misled by unrelated tasks.  This carefully designed synthetic prior is what enables the MTPFNs to achieve robust transfer learning. We can see the impact of this data generation procedure in Figure 3: when we train the MTPFN with varying proportions of unrelated tasks $p$, we find that the MTPFN is *not* robust to negative transfer when none of the tasks are unrelated $(p = 0)$.
>
> > If real-world tasks and their relationships do not resemble this mixture prior, the model’s performance may degrade
>
> Our empirical results demonstrate **consistent competitive performance across diverse benchmarks**. In our empirical evaluation, we find that MTPFNs are able to confidently leverage relevant information from related tasks while avoiding negative transfer from unrelated ones. Furthermore, although the value of $p$ is fixed during training, through the millions of generated synthetic datasets that the model is trained on, the model still sees many instances where all of the source tasks are unrelated, none of the source tasks are unrelated, and everything in between. With this data generation procedure, we expect the model to adapt fairly well at test-time to any number of unrelated tasks.
>
> We also include sensitivity analyses in Appendix A, where we vary the synthetic data generation process and benchmark how different PFNs perform across our diverse set of real-world tasks.
>
> With that said, the question of what data generating procedure is “optimal” remains an open and important research direction, and we hope that our paper inspires further exploration in this area. For instance, priors inspired by LMC models could capture more complex inter-task relationships, while priors which incorporate low-rank task relationships could also provide alternative useful inductive biases. We believe our work provides a solid foundation for exploring these richer prior generation procedures.
>
> > The paper does not address scalability wrt input dimensionality
>
> We acknowledge that scaling multitask PFNs to high-dimensional inputs is an important research direction. Our current method focuses on benchmarks where the main challenge is on identifying shared structures and effectively learning inter-task relationships rather than handling extremely large input spaces. There have been recent works explicitly addressing high-dimensional PFNs [e.g. 1]. These techniques could in principle be layered on top of our multi-task approach, and exploring these extensions is an interesting area of future work.
>
> [1] Kolberg et al, TabPFN-Wide: Continued Pre-Training for Extreme Feature Counts
>
> > Comparisons with other fine-tuning approaches
>
> There are many modern fine-tuning methods, and it’s likely that they would lead to slightly different empirical outcomes! However, fine-tuning typically requires substantially more data and compute to perform well, and also generally adapts a model to perform well in one specific domain. For instance, for a model pre-trained on Logistic Regression, it’s possible that fine-tuning the model with LoRA on SVM would enable it to perform well on SVM and Logistic Regression, but it would still not be able to generalize to other domains such as GP draws. In contrast, multi-task in-context learning enables a single model to perform effectively across multiple tasks without gradient updates. This may be particularly valuable in data-scarce regimes, where fine-tuning may be impractical.
>
> > Hierarchical attention mechanism is very dense
>
> Thanks for the feedback! We needed to condense the explanations due to space constraints, but with the additional page allotted for the rebuttal, we have included many more details in the updated pdf. We have also added a new experiment for the rebuttal (in Figure 9) which further demonstrates the benefits of the hierarchical attention mechanism.

---

### Official Review · Reviewer_yMBa · 2025-11-06

**Soundness:** 2
**Presentation:** 3
**Contribution:** 2
**Rating:** 4
**Confidence:** 3

**Summary:**

The paper introduces multi-task prior-data fitted networks (MTPFNs) for the problem of Bayesian optimization (BO) from a transfer learning perspective. In particular, the work raises concerns about how the intrinsic corregionalization model (ICM) usually considered for the mixing structure of multi-output GPs may struggle when there is a negative influence between output variables or transfer undesired modelling properties are transferred. From the perspective of the authors, an PFN with a categorical encoding of tasks is more robust than ICM and the usual multi-task GP that practitioners. The paper dedicates a considerable amount of text and some empirical results to prove this hypothesis and to show that MTPFNs are superior for these cases. Finally, the proposed method is tested on a couple of hyperparameter-fitting benchmarks, summarised in Figure 6 (pp. 9) of the manuscript.

**Strengths:**

- One of the main hypotheses of the paper, which is repeated several times throughout the manuscript, is that multi-task Gaussian processes make strong assumptions about the relationship between tasks, which sometimes degrades performance when such assumptions are violated. Interestingly, I agree with this hypothesis --- and I consider it valid; however we should also mention that there is always a trade-off between "flexibility" and "optimization". It is obvious that GPs, with their usual linear structure, impose certain strong assumptions that NNs don't. But we must also be aware that such introduction of NNs is not always beneficial, particularly in small data uncertainty-based problems. I do think more or less everyone in the community agrees on that.
- The work is thorough and has a good scientific spirit in the sense of explaining what the motivations are behind introducing MTPFNs and what they are competing with. In general, it is well written and perfectly clear, no doubts on that.
- I miss perhaps some broader perspective on the BO's SOTA methods and baselines, like for instance going deeper into that literature and motivating better the problem of multi-task models for BO. Still, I do think the paper did a good job on including critical references and overall on those around PFNs and ICM. The empirical results and the testing on the BO hyperparameter benchmark are sufficient, but not as complete as one might expect for this sort of venue and submission.

**Weaknesses:**

- To me, the central concerns and problems faced in the paper revolve around the "negative influence/transfer" of some tasks while performing inference. There are a lot of claims about the impossibility of MTGPs and ICM to address this problem, while the categorical encoding of tasks in the PFN can do that. In this long discussion, I miss a lot of the math, honestly. The ICM linear model is barely mentioned, and its modelling choices are not shown in the form of an equation. With this, even for someone with some experience with MOGPs + ICM, LCM, or convolutions, it is difficult to follow some claims made in the paper. I would be happy to engage in the discussion with the authors, because I really want to understand where this hypothesis comes from and know more about it.
- Going a bit deeper into the technical details and these limitations of the ICM model, I do think that the claims made from the performance shown in Figure 1 are missing numbers, like predictive posterior on test data etc. Otherwise is difficult to judge visually. It is interesting that authors are somehow indicating that the ICM is not able to reduce the contribution of the (periodic and unrelated) task; let's say that the mixing linear coefficient should be near zero. But I am not entirely convinced that the MTPFN is superior here. The red task has a clear periodic structure that the MTPFN is not using in a horrible way for the performance; it is just showing uncertainty, who knows which of the lengthscales influenced by the green tasks. Additionally, the imbalance in the number of observations between tasks affects somehow, so more empirical tests should be considered for better comprehension.
- Beating ICM and MTGP final results should at least be shown, and we should also have (as readers/reviewers) some notion of what the computational bill of doing this transition. BO is particularly well known for being applied to
- heavy problems, but additional evidence on time consumption, parameters, etc, should be included. I am aware of the notions provided in Section 3.2.1, but they are not very general to understand the final procedure.
- Last but not least, I am somehow concerned that the focus of the paper is BO, but it's empirical validation is limited and somehow the general BO mission does not shine a lot in the manuscript, which gives the work an odd communication thread.

**Questions:**

- Section 3.2 is a bit difficult to understand, in the sense that two techniques are introduced without much context (for someone very familiar with PFNs maybe is easier) and later the hierarchical attention mechanism. From my point of view I can see that they are different versions of the model and they improve it on different limitations. But later for Section 4 is not clear to me what is considered, what is not, how they related to the benchmarks and the data-based problem faced. I would like to hear more on that.
- This is a very-minor point, but I do think a lot of works and interesting references could also be added to guide the reader through the BO problem. Garnett (2023) as a summary of Bayesian optimization is a bit limited to me. I'm pretty sure that a lot of cool contributions came out before that book ;))

---

> ### Author Response · Authors · 2025-11-24
> **Response [1/2]**
>
> Thank you for your thoughtful review! We address your points and questions below, and we have also updated the pdf with significantly more detail regarding the related work and our proposed method.
>
> >  Understanding why ICM suffers from negative transfer
>
> The ICM is a widely used MTGP which models each task as a scaled version of a shared latent function: $f(x, t) = a_t u(x)$ where $u(x) \sim \mathcal{GP}(0, k_\text{inputs}(x, x’)$ and $a_t$ is a task-specific coefficient. The resulting covariance between input $x$ from task $t$ and input $x’$ from task $t’$ is then $k((x, t), (x', t')) = a_t a_t’ k_\text{inputs}(x, x') = k_\text{inputs}(x, x') \cdot k_\text{tasks}(t, t').$
> Crucially, the same input kernel $k_\text{inputs}$ is shared across all of the tasks, forcing them to all have the same lengthscale. When this assumption is broken, and some tasks have different lengthscales than other tasks, then the model may suffer from negative transfer. This can be seen in the visualization of Figure 1: the two green source tasks share the same lengthscale as the target task, but the last red task has a completely different lengthscale / smoothness compared to the target task. Because of this one unrelated task, the ICM learns the short lengthscale, even though the target task and source tasks all seem to have long lengthscales, and so the ICM has high uncertainty and is less able to capture the properties of the target task compared to the MTPFN.
>
> Other MTGPs such as LMC do not suffer from these same assumptions. The LMC expresses each task as a linear combination of independent GPs $f(x, t) = \sum_{q=1}^Q a_qt u_q(x)$, where $u(x) \sim \mathcal{GP}(0, k_q(x, x’)$ and $a_qt$ are the coregionalization coefficients. LMC models (with Q>1) can represent complex relationships and different lengthscales per latent kernel; however, these models come with higher computational complexity and may lead to overfitting [1, 2].
>
> [1] Alvarez and Lawrence, Efficient Convolved Multiple Output Gaussian Processes  \
> [2] Dai et al, Efficient Modeling of Latent Information in Supervised Learning using Gaussian Processes
>
> We have added this discussion to the background section of our paper.
>
> > Claims in Figure 1 are missing numbers
>
> Figure 1 is meant to be an illustrative example of how negative transfer can occur. Models such as the ICM may learn a short lengthscale because of the unrelated source task, and therefore have unnecessarily high levels of uncertainty. across the input space. Ensemble models such as ScaML may also pick up on the short lengthscale and make poor predictions. TabPFN, which was trained using a different data-generation procedure, does not perform as well as the MTPFN, which was explicitly trained for the multi-task transfer learning setting. Finally, we see that the two related source tasks do provide relevant information: the MTPFN, which has access to all four tasks, outperforms models which only have access to the target task.
>
> We have updated the figure to include the negative log-likelihoods for each model, and we see that MTPFNs do in fact achieve lower losses in this setting.
>
> > We must be aware that the introduction of NNs is not always beneficial, particularly in small data uncertainty-based problems. I do think more or less everyone in the community agrees on that.
>
> We would like to clarify that our approach does not train a neural network from scratch on limited data. Instead, the PFN is trained entirely on synthetic data generated from a prior. For instance, if the synthetic data is generated through draws from a GP, then the PFN will learn to replicate the GP’s inductive biases and uncertainty estimates [1]. Some even argue that PFNs will replace GP-based approaches even for small-data settings [2]. In our work, we base our synthetic data generation procedure on the ICM, which allows the PFN to capture task correlations and provide well-calibrated uncertainty estimates even in small-data settings.
>
> [1] Müller et al, Transformers Can Do Bayesian Inference \
> [2] Müller et al, Position: The Future of Bayesian Prediction Is Prior-Fitted
>
> > ICM and MTGP final results should be shown
>
> We report the ICM results (which is a commonly-used MTGP model) in all of our benchmarks. Could you elaborate on what you mean by this statement?

---

> ### Author Response · Authors · 2025-11-24
> **Response [2/2]**
>
> > The general BO mission does not shine a lot in the manuscript.
>
> Our work is explicitly framed around transfer learning within BO, not just multi-task learning. While multi-task methods focus on learning the relationships between tasks, the objective of BO is to make efficient decisions under uncertainty. Our data generation procedure is designed for the PFN to identify the patterns which are useful for iterative BO setting, rather than only improving predictive accuracy across multiple tasks.
>
> Furthermore, many papers in BO literature focus on designing effective surrogate models (e.g. ScaML) without questioning whether this counts as “doing BO”. Our work follows the same paradigm, where the PFN is a learned surrogate model trained to produce useful posterior predictive distributions which are then used for sequential decision-making.
>
>
> > Additional evidence on time consumption, parameters, etc should be included
>
> As with all PFN-type methods, there is an initial one-time training cost incurred by us. In our case, we trained our models for 24 hours on an NVIDIA V100. However, after this one-time training, the model can be repeatedly applied with very little cost. Furthermore, we will release the model weights, so future practitioners will be able to directly use the model without their own pre-training.
>
> After this initial cost, the inference time of PFNs is significantly faster than GP-based methods. As we demonstrate in Figure B.2 in the supplementary material, MTPFNs are able to perform inference on an order of magnitude more data compared to traditional GP methods, with joint-modeling methods such as ICM taking up to 12 seconds to fit 300 samples ensemble-based methods such as ScaML taking up to 2 seconds, and the MTPFN taking only 0.02 seconds. Furthermore, we see in Figure 4 (right) that MTPFNs are comparable to NUTS sampling; however, the NUTS algorithm can take upwards of 10 minutes per iteration; meanwhile, the MTPFN is able to perform inference in under 1 second (~600x speedup).
>
> > Section 3.2 is a bit difficult to understand
>
> Thank you for the feedback! We needed to condense the explanations due to space constraints, but with the additional page allotted for the rebuttal, we have updated the main text to be more explicit about the hierarchical attention mechanism and how it compares to alternative task encoding methods. We have also added a new experiment for the rebuttal (in Figure 9) which further demonstrates the benefits of the hierarchical attention mechanism.
>
> > I think a lot of works and interesting references could be added to guide the reader through the BO problem.
>
> We agree that there’s been lots of interesting work in this space! We have added further discussions to our related works.
>
> We have put substantial effort into addressing your questions and improving the clarity of our manuscript, and we hope you will consider updating your score accordingly.

---

### Meta-Review · Area_Chair_qXEU · 2026-01-03

**Summary:**

This paper proposes a multi-task version of PFN to perform better transfer learning for Bayesian optimization.

The concerns from reviewers centered on the insufficient evidences to support the claims (e.g., the proposed method excels at dealing with negative transfer but most traditional MTGPs lack the ability; MTPFNs are very good for Bayesian optimization etc), incremental novelty, clarity and assumptions. While the authors addressed a number of the issues in the rebuttal and updated paper, some concerns remain not fully addressed, such as the lack of results on Bayesian optimization, limited practicality regarding the assumptions related to in-distribution data/synthetic prior and insufficient discussions on results (like Figs 4 and 5).

On point stood out in the rebuttal is the authors' position on whether the focus is BO or purely designing effective surrogate models. The authors' response to Reviewer yMBa seems to have contradictory goals, claiming both the approach is "designed for the PFN to identify the patterns which are useful for iterative BO setting, rather than only improving predictive accuracy" and that this paper "follows the same paradigm" of  designing effective surrogate models without questioning whether this counts as “doing BO”. If the authors insist the goal is to obtain better surrogate model and BO is only an application, the paper needs to be re-written significantly. Judging from the paper itself, the goal is BO, so I do think more SOTA BO baselines and comprehensive evaluations on BO tasks should be performed to demonstrate the effectiveness of the proposed method. Moreover, in the rebuttal, the authors argued that the empirical results have shown that the assumptions (in-distribution/synthetic prior) were okay to make, but it seems to me that the current experiments with limited baselines and realistic tasks do not sufficiently support those arguments.

For the reasons above, I recommend rejection.

**Reviewer Concerns:**

Many problems regarding the clarity of the proposed method were properly addressed.

Reviewer yMBa's concern on BO was not addressed. Reviewer dMBL's concerns on the lack of discussions of results and how well the method works for a suite of tasks (so that it's not by luck) were not addressed.

Reviewer f9Ye and dMBL both had concerns on the in-distribution assumption. Reviewer gndA had a similar concern about the synthetic prior. The authors argued that the empirical results have shown the practicality, but I think the results are relatively limited.

**Reviewer Scores:**

I believe that the reviewers will not increase the scores since their concern was not properly addressed as explained above.

---

### Decision · Program_Chairs · 2026-01-26

Reject